# Tumor-Associated Trypsin Inhibitor (TATI) as a Biomarker of Poor Prognosis in Oropharyngeal Squamous Cell Carcinoma Irrespective of HPV Status

**DOI:** 10.3390/cancers13112811

**Published:** 2021-06-04

**Authors:** Anni Sjöblom, Ulf-Håkan Stenman, Jaana Hagström, Lauri Jouhi, Caj Haglund, Stina Syrjänen, Petri Mattila, Antti Mäkitie, Timo Carpén

**Affiliations:** 1Department of Pathology, University of Helsinki and HUS Helsinki University Hospital, P.O. Box 21, FI-00014 Helsinki, Finland; jaana.hagstrom@hus.fi (J.H.); timo.carpen@fimnet.fi (T.C.); 2Department of Clinical Chemistry, University of Helsinki and HUS Helsinki University Hospital, P.O. Box 63, FI-00014 Helsinki, Finland; ulf-hakan.stenman@pp.fimnet.fi; 3Research Programs Unit, Translational Cancer Biology, University of Helsinki, P.O. Box 63, FI-00014 Helsinki, Finland; caj.haglund@helsinki.fi; 4Department of Oral Pathology and Oral Radiology, University of Turku, Lemminkäisenkatu 2, FI-20520 Turku, Finland; stisyr@utu.fi; 5Department of Otorhinolaryngology—Head and Neck Surgery, University of Helsinki and HUS Helsinki University Hospital, P.O. Box 263, FI-00029 Helsinki, Finland; lauri.jouhi@helsinki.fi (L.J.); petri.mattila@hus.fi (P.M.); antti.makitie@helsinki.fi (A.M.); 6Department of Surgery, University of Helsinki and HUS Helsinki University Hospital, P.O. Box 440, FI-00029 Helsinki, Finland; 7Department of Pathology, Turku University Hospital, Kiinamyllynkatu 10, FI-20520 Turku, Finland; 8Division of Ear, Nose and Throat Diseases, Department of Clinical Sciences, Intervention and Technology, Karolinska Institutet and Karolinska Hospital, SE-171 76 Stockholm, Sweden; 9Research Program in Systems Oncology, Faculty of Medicine, University of Helsinki, P.O. Box 63, FI-00014 Helsinki, Finland

**Keywords:** OPSCC, HPV, TATI, survival

## Abstract

**Simple Summary:**

Oropharyngeal squamous cell carcinoma (OPSCC) is a form of head and neck cancer in which human papillomavirus (HPV) infection has been shown to play a major role in disease development. The survival rates of HPV-positive patients are favorable compared to HPV-negative patients, but the reason for this phenomenon remains unclear. The management of OPSCC is complex, and development of novel treatment options is urgently required. Various possible factors affecting survival have been explored, including the tumor environment and cancer-related proteases. Our aim was to study a protease inhibitor known as tumor-associated trypsin inhibitor and its correlation with survival and clinical data in OPSCC patients.

**Abstract:**

Background: We studied the role of tumor-associated trypsin inhibitor (TATI) in serum and in tumor tissues among human papillomavirus (HPV)-positive and HPV-negative OPSCC patients. Materials and methods: The study cohort included 90 OPSCC patients treated at the Helsinki University Hospital (HUS), Helsinki, Finland, in 2012–2016. TATI serum concentrations (S-TATIs) were determined by an immunofluorometric assay. Immunostaining was used to assess tissue expression. HPV status was determined with a combination of p16 immunohistochemistry and HPV DNA PCR genotyping. The survival endpoints were overall survival (OS) and disease-specific survival (DSS). Results: A significant correlation was found between S-TATI positivity and poor OS (*p* < 0.001) and DSS (*p* = 0.04) in all patients. In HPV-negative cases, S-TATI positivity was linked to poor OS (*p* = 0.01) and DSS (*p* = 0.05). In HPV-positive disease, S-TATI positivity correlated with poor DSS (*p* = 0.01). S-TATI positivity was strongly associated with HPV negativity. TATI serum was negatively linked to a lower cancer stage. TATI expression in peritumoral lymphocytes was associated with favorable OS (*p* < 0.025) and HPV positivity. TATI expression in tumor and in peritumoral lymphocytes correlated with lower cancer stages. Conclusion: Our results suggest that S-TATI positivity may be a biomarker of poor prognosis in both HPV-positive and HPV-negative OPSCC.

## 1. Introduction

The incidence of OPSCC has been increasing in recent years, particularly in Western countries [1]. Although over 100,000 new cases of oropharyngeal cancers are diagnosed yearly worldwide, the majority is diagnosed in developed countries [2,3]. Squamous cell carcinomas form over 90% of the newly diagnosed oropharyngeal cancers [2,3,4,5,6]. The median overall 5-year survival in Finland is 64% among men and 70% among women [7]. The most significant risk factors for OPSCC are smoking, heavy alcohol use, and HPV infection. Today, more than half of the new OPSCC cases are associated with HPV [8] and HPV-positive OPSCC and HPV-negative OPSCC are considered as separate disease entities. In HPV-positive OPSCC, the symptom profile and tumor characteristics differ from HPV-negative OPSCC, and the treatment response and the prognosis are usually substantially more favorable in HPV-related disease [9,10,11]. The prognosis for HPV-negative OPSCC remains relatively poor [12,13], and the explanation for poor survival rates remains unclear. To improve treatments and diagnostics, it is important to discover novel information on previously undiscovered prognostic factors and mechanisms affecting survival.

Various biomarkers have previously been associated with OPSCC, such as p16 [9,14,15]. Other potential biomarkers with possible prognostic value in OPSCC include Cyclin D1 and matrix metalloproteinases 1 and 2 [16,17,18,19]. Furthermore, recent studies have shown that HPV16 E6 and E7 serum antibody levels may predict OPSCC in advance [20,21]. However, to our knowledge there are currently no prognostic biomarkers or diagnostic serological biomarkers that are used for individualization of treatments for head and neck squamous cell cancers. Thus, further research on promising biomarkers is warranted. This study is focused on a biomarker known as tumor-associated trypsin inhibitor (TATI), which is associated with multiple malignancies, but has rarely been studied in oropharyngeal cancer [22,23].

TATI, also known as pancreatic secretory trypsin inhibitor (PSTI) or serine peptidase inhibitor Kazal 1 type (SPINK1), is a trypsin inhibitor that functions mainly in the pancreas, where it serves as a suppressor of premature trypsinogen activation [24]. The mechanisms regulating extrapancreatic TATI secretion are only partially known. The reference range of TATI concentration in serum (S-TATI) is 3.2–16 μg/L in healthy individuals [25]. S-TATI has been shown to increase in several non-malignant conditions, such as acute pancreatitis and various other severe inflammatory diseases. Elevated secretion of TATI in cancer patients was first found in the urine of patients with ovarian cancer [26,27] and it has since been detected in serum and tumor tissue in various malignancies [25].

In addition to the role if TATI as a diagnostic tumor marker, it may have value as a prognostic marker [28], and as a target for cancer treatment [29]. The purpose of this study was to evaluate the significance of TATI as a biomarker in HPV-positive and HPV-negative OPSCC based on its expression in cancer tissue and its serum concentrations and to investigate its value as a prognostic factor.

## 2. Materials and Methods

### 2.1. Study Population

This study was based on an existing database used in previous studies [30,31] and it is consisted of a cohort of 224 patients with newly diagnosed oropharyngeal squamous cell carcinoma at the HUS, Helsinki, Finland in 2012–2016.

Inclusion criteria for this study were a prospectively collected serum sample analyzed for TATI concentration, and the previously determined HPV status in tissue samples. Altogether, 90 of the 224 patients fulfilled these criteria and were included in the final analysis.

Data were collected from electronic patient records and were updated during the study for assessment of prognosis. The data included clinical characteristics such as age, gender, smoking, use of alcohol, TNM class, stage, grade of differentiation, tumor localization, HPV status using an algorithm described by Smeets et al. [32] As a treatment modality, patients received either (chemo)radiotherapy or surgery with or without post-operative (chemo)radiotherapy. All patients were treated with curative intent. The median follow-up time was 47.5 months (range 0.0–60.0). The survival endpoints were overall survival (OS) and disease-specific survival (DSS).

The study was approved by the Research Ethics Board at the HUS and an institutional permission for the study was granted (Dnr: 51/13/03/02/2013).

### 2.2. Tissue Microarrays

Tissue microarray (TMA) blocks were prepared from primary tumors with the assistance from digital software by Auria Biobank (Turku, Finland). Representative areas were selected from hematoxylin and eosin-stained slides and six 1-mm thick core biopsies from each tumor were detached from paraffin blocks. The cores were then placed in another block with a semiautomatic tissue microarrayer (Beecher Instruments, Silver Spring, MD, USA).

### 2.3. HPV Status Determination

In our study, HPV status was determined by Smeets’s algorithm using a combination of p16 immunohistochemistry (IHC) and HPV DNA detection.

All the paraffin-embedded samples were immunostained with p16-INK4a antibody. A positive control was included. HPV expression was considered as positive in samples where more than 70% of the tumor cells were positive. 

For HPV DNA detection, DNA was first extracted from the tumor tissue samples followed by PCR-based genotyping using a Multiplex HPV Genotyping kit^®^ (DiaMex GmbH, Heidelberg, Germany) as previously described [30]. Positivity of the high-risk (hr) HPV genotypes (subtypes 16, 18, 31, 33, 35, 39, 45, 51, 52, 56, 58, and 66) in the samples was considered as a positive result for hrHPV DNA in the tumor. 

The methodology of these procedures has been described in detail previously [30,31,33].

### 2.4. Determination of TATI Serum Concentrations

All serum samples (*n* = 90) were collected prior to treatment. S-TATI was determined by a time-resolved immunofluorometric assay (IFMA) using monoclonal antibodies (MAbs) produced in-house [34]. The capture antibody was coated onto microtitration wells and the detector antibody labeled with a europium chelate. TATI purified from the urine of a patient with ovarian cancer was used as a calibrator [27]. The calibrators covered the concentration range of 0.5–150 µg/L. The sample volume was 25 µg/L and the total assay volume was 200 µL. The detection limit of the assay was 0.15 µg/L and the CV < 10% was at concentrations in the range of 1–50 µg/L [34]. The reference range for S-TATI was 3.2–16 µg/L and it was determined based on the central 95% reference interval in a group of 152 apparently healthy subjects. The lower reference limit was 3.1 µg/L and the upper reference limit was 16 µg/L [34]. The upper reference limit was lower than that for the initially used radioimmunoassay (21 µg/L) [27]. This was attributed to the higher non-specific background of the RIA. Samples, where S-TATI exceeded the selected cut-off value 17 µg/L [24,25,35,36] were considered S-TATI positive.

### 2.5. IHC of TATI

IHC staining of TATI was analyzed in TMA slides. Deparaffinization and rehydration of the tissue slides was performed with Sakura Tissue-Tek DRS. The HIER method (heat induced epitope retrieval) was performed with the Pretreatment Module, Agilent Dako (Dako Denmark Aps, Glostrup, Denmark) to improve antigen retrieval in the samples. Endogenous peroxidase blocking was performed with EnVision Flex peroxidase-blocking reagent (Dako). A monoclonal TATI antibody (MAb 6E8) [37] was used as the primary antibody. Dako REAL Antibody Diluent S2022 (Dako) was used for antibody dilutation. EnVision Flex/HRP SM802 DM827 (Dako) was used as a secondary antibody. The chromogen was EnVision Flex DAB (Dako). Hematoxylin was used for counterstaining. Staining was performed with Autostainer 480 (Thermo Fisher Scientific, Vantaa, Finland). After the staining procedure, the specimen was dehydrated and then mounted with Pertex Histolab mounting media (Histolab Products Oy, Gothenburg, Sweden). We have not found evidence of non-specific staining while previously applying this method for TATI IHC and the specificity of the TATI antibody is described in earlier studies [34,36,37].

### 2.6. Sample Scoring

Scoring of TMA blocks was performed by two researchers (Anni Sjöblom and Jaana Hagström). A consensus was achieved in case of disagreements. TATI expression was assessed in the tumor tissue as well as in the peritumoral lymphocytes. The scoring of TATI in the samples was graded as described in Appendix A.

### 2.7. Data Analysis

Data were collected and analyzed with IBM SPSS Statistics software program version 25. S-TATI and TATI expression in tumor tissues were compared with age, gender, smoking status, alcohol use, TNM class, stage, histological grade, tumor site and HPV status. Crosstab comparisons were performed with the χ^2^-test and for normally distributed continuous variables, independent sample *t* test was used. For the survival analysis, the selected endpoints were 5-year OS and DSS. OS was defined as the time from the last day of treatment to death from any cause. DSS was defined as the time from the last day of treatment to the date of OPSCC-related death. OS and DSS were assessed with the log-rank-test and illustrated with the Kaplan–Meier-estimator using GraphPad Prism- software version 9. For the univariable and multivariable analysis, the Cox regression analysis was performed. TATI serum concentrations were logarithmically transformed to obtain normal distribution. Variables receiving *p*-values under 0.05 in the univariable analysis were included to the multivariable analysis.

## 3. Results

### 3.1. TATI Serum Concentrations and Clinical Characteristics

S-TATI was determined in 90 serum samples. Based on a selected cut-off value of 17 µg/L, 21 (23.3%) of the patients were considered TATI positive and 69 (76.7%) S-TATI negative. The crosstab comparisons of S-TATI according to clinical parameters are presented in Table 1.

The correlation between S-TATI and HPV status was statistically significant (*p* < 0.001). Most (90.60%) of the HPV-positive patients were S-TATI negative. Furthermore, S-TATI negativity was linked to lower cancer stage and higher histological grade (82.5% of the patients had cancer stages I-II and 83.3% of the patients were grade III). In addition, S-TATI negativity correlated with tonsil as tumor site (90.6% of the patients).

A majority (96.4%) of the non-smokers were S-TATI negative. Additionally, most (83.3%) of the former smokers were S-TATI negative. Among the smokers, 53.3% were S-TATI negative and 46.6% were S-TATI positive.

S-TATI positive patients had reduced OS during the 5-year follow-up and S-TATI positivity was linked to poor OS (*p* < 0.001) and DSS (*p* = 0.04) in the whole cohort. Furthermore, S-TATI positivity correlated with poor OS (*p* = 0.01) and DSS (*p* = 0.05) in HPV-negative patients and with poor DSS (*p* = 0.01) in HPV-positive patients. The survival curves according to S-TATI and OS are presented in Figure 1.

In the multivariate Cox regression analysis, poorer OS correlated significantly with age (adjusted Hazard ratio (HR) 1.08, 95% Confidence interval (CI) 0.03–1.13, *p* = 0.004) and S-TATI (adjusted HR 2.47, 95% CI 1.26–4.84, *p* = 0.009). In addition, poorer DSS was associated with age (adjusted HR 1.07, 95% CI 0.01–1.14, *p* = 0.018) and S-TATI (adjusted HR 2.54, 95% CI 1.07–6.02, *p* = 0.034) in the Cox regression multivariate model. The results of the multivariable analysis are presented in Table 2. In the multivariate Cox regression analysis with p16 as a separate variable, S-TATI correlated significantly with OS (adjusted HR 2.49, 95% CI 1.28–4.83, *p* = 0.007) and DSS (adjusted HR 2.54, 95% CI 1.09–5.94, *p* = 0.031) and the results are presented in Appendix A. 

### 3.2. TATI Immunoexpression and Clinical Characteristics 

Both serum and tissue data were available for 90 patients. For TATI IHC, adequate samples with HPV status determination were available for 77 (85.6%) patients. TATI was assessed in the tumor tissue in all 77 samples and in the tumor-adjacent lymphocytes in 76 samples (Figure 2a–c). Most of the samples showed moderate or strong IHC staining of TATI in both the tumor (63.3%) and in the peritumoral lymphocytes (60.5%). TATI-immunopositivity was cytoplasmic. TATI expression in IHC was associated with certain patient characteristics but not with S-TATI. Crosstab comparisons of TATI IHC and clinical characteristics are presented in Table 3 and Table 4.

Moderate or strong expression of TATI in tumor tissue and in peritumoral lymphocytes (Figure 2a,b) was observed in 83% and 69.8% of patients with stage I–II disease, respectively. Expression of TATI in tumor tissue did not significantly correlate with other clinical parameters. However, elevated expression of TATI in peritumoral lymphocytes was linked to HPV positivity (66.6%) and lower T class (71.4%). In addition, patients with moderate or strong expression of TATI in peritumoral lymphocytes had a favorable OS (*p* = 0.025). However, this result was only observed in the whole patient cohort and the comparisons between HPV status and survival was not statistically significant. No correlation was seen between DSS and TATI tissue expression.

## 4. Discussion

Our study revealed novel information on TATI in both serum and in tumor tissue among OPSCC patients. Our findings suggest that S-TATI positivity is linked with poor prognosis of OPSCC irrespective of HPV status. In recurrent HPV-negative disease managed with primary definitive oncological treatment, the prognosis is usually considerably impaired. Therefore, determining the S-TATI could provide an advantage for the patient when establishing the treatment plan. Additionally, S-TATI determination could be valuable in monitoring purposes during posttreatment follow-up. In earlier studies TATI was shown to have prognostic value in other cancers [35,38]. Considering our results, further research with larger cohorts assessing TATI as a prognostic biomarker in OPSCC is warranted. 

To the best of our knowledge, there are no studies on TATI in OPSCC. Interestingly, we found that S-TATI was associated with survival in both HPV-negative and HPV-positive disease, particularly in HPV-negative OPSCC. Given the limitations of the small sample size, we were not able to thoroughly analyze the effect of HPV status on S-TATI levels. As S-TATI positivity was associated with HPV negativity, it is likely that the association between poor prognosis and S-TATI positivity in the whole OPSCC group is strongly related to concurrent HPV negativity. Interestingly, we found a correlation between S-TATI positivity and poor DSS in HPV-positive patients. According to these findings, S-TATI positivity may have individual effects on survival unrelated to HPV negativity, although our sample size in this analysis was small. Studies on larger cohorts are needed to determine the role of HPV status and S-TATI for prognostication purposes. However, we observed that elevated S-TATI appears to function as an independent prognostic biomarker in OPSCC (Table 2). Similar results were observed in other malignancies [28,39,40].

In our study, previous and present smoking correlated strongly with S-TATI positivity (*p* < 0.001). Interestingly, similar results were observed in a previous study [41]. Furthermore, evidence of possibly smoking-related mutations in TATI has been found in chronic pancreatitis [42]. We speculate that smoking could increase mutations in TATI, possibly impairing its functions and thus leading to poor prognosis, TATI upregulation in serum, or both. Further research is required to establish the connection between smoking and TATI upregulation in OPSCC.

The mechanisms regulating TATI secretion in cancer are not known. Thus, the cause of the elevated TATI levels in serum and tissue in OPSCC and other malignancies is unclear. In general, it appears that TATI may have different effects on prognosis and pathogenesis depending on the type of malignancy [24]. There are several theories concerning the regulation of TATI in cancer and one of most frequently encountered is the connection of TATI with the protease tumor-associated trypsin (TAT) [43,44,45]. TAT and TATI are often produced simultaneously, and TAT has previously been linked to several malignancies and pathological conditions [46]. We speculate that TATI is acting through TAT, which is known to activate other proteinases mediating tissue destruction and invasion [43,47]. Low tissue expression of TATI could increase TAT activity due to the lack of inhibition. This phenomenon may facilitate local tumor invasion, which is associated with poor prognosis. This theory would support our findings of S-TATI positivity signaling impaired survival but does not align with our IHC results. Overexpression of TAT along with low TATI expression has previously been associated with enhanced tumor-cell invasion in oral and gastric cancer [47]. These results conflict with our findings regarding S-TATI positivity linked with poor prognosis. This contradiction might reflect the fact that their study was done in vitro with cell lines and is not directly comparable with our clinical study. 

Recent studies on other malignancies have provided compelling theories on TATI and its effects on cancer aggressiveness and survival. The importance of the tumor microenvironment and the tumorigenic abilities of stromal cells is associated with prognosis [29,45]. The senescence-associated secretory phenotype (SASP), a known feature of cellular senescence, is associated with cancer and stromal cells [48]. It has been suggested that SASP in stromal cells is a mediator of paracrine S-TATI upregulation and of various cancer-promoting mechanisms (such as invasiveness), and it is associated with poor prognosis in prostate cancer [29]. Although SASP was observed after chemoradiation by Chen et al., similar events might stimulate S-TATI expression in OPSCC. Furthermore, SASP has previously been shown to trigger various cancer-promoting features with different mechanisms in oral squamous cell carcinomas [49]. Therefore, we speculate that SASP may be an important contributor to S-TATI upregulation in OPSCC and these events may also lead to poor prognosis as well. Our findings of S-TATI positivity being linked to poor prognosis support this theory. These explanations do not completely align with our IHC results.

Although we did not observe any correlation between S-TATI and TATI expression in tumor tissues, there was a moderate correlation between S-TATI negativity and tissue positivity. Interestingly, it has not been possible to establish an association between elevated S-TATI and tumor tissue in previous studies [28,50]. It has been speculated that TATI protects tissues against cancer invasion [43]. This would be consistent with our observations of TATI positivity of tumor-adjacent lymphocytes and several favorable associated factors, i.e., lower T class and stage. We speculate that while TATI expression is mainly increased in tumors and their surrounding tissues, TATI protein possibly locates in the tissues, which might reduce secretion into the blood. According to our results both negative S-TATI and high TATI expression in the peritumoral lymphocytes were linked to lower disease stage.

Several previous studies have failed to show elevated S-TATI and TATI expression in tumor tissues as associated prognostic factors [43,44,50]. Our findings indicate a significant association between poor prognosis and S-TATI positivity and these results appear to be more distinct compared to the IHC results. Further research is required to establish correlation between TATI expression in tissues and S-TATI as prognostic factors in cancer. 

The present results could provide a novel path for future OPSCC research. To the best of our knowledge, there are no other serum biomarker assays for diagnostic and monitoring purposes of oral and oropharyngeal cancers.

## 5. Conclusions

Based on the present results, serum TATI appears to be a promising prognostic biomarker in HPV-positive and HPV-negative OPSCC. Prospective studies and further research in larger cohorts are warranted to determine the functions and physiology of TATI in OPSCC more thoroughly.

## Figures and Tables

**Figure 1 cancers-13-02811-f001:**
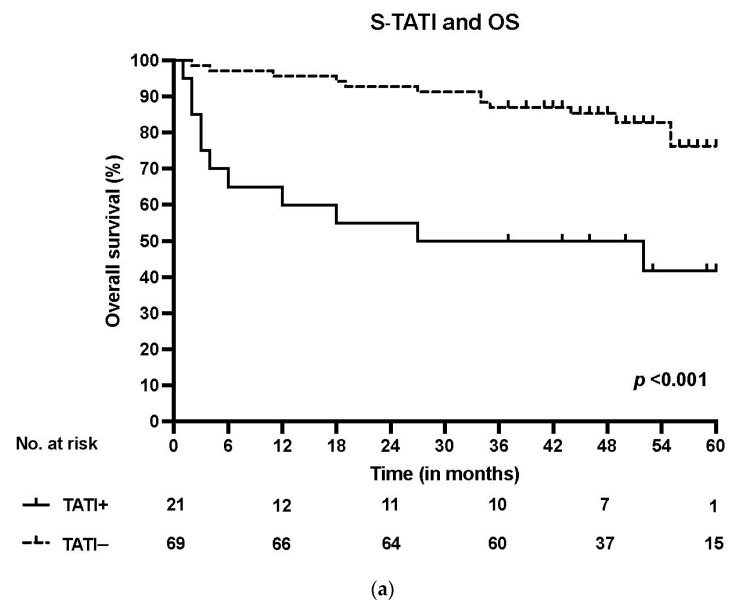
(**a**) Overall survival (OS) according to positive (>17 µg/L) and negative (<17 µg/L) S-TATI in the whole patient cohort. (**b**) Overall survival (OS) according to positive and negative S-TATI in HPV-positive OPSCC patients. (**c**) Overall survival (OS) according to positive and negative S-TATI in HPV-negative OPSCC patients.

**Figure 2 cancers-13-02811-f002:**
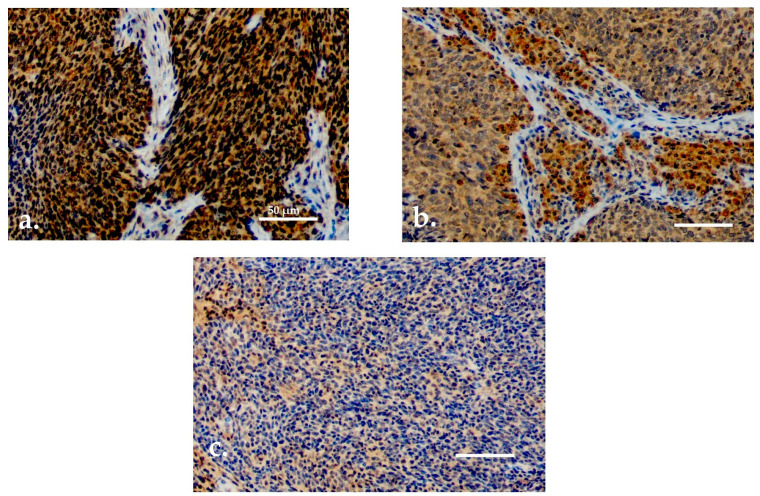
(**a**) Moderate/strong TATI expression in tumor tissue. (**b**) Moderate/strong TATI expression in peritumoral lymphocytes and mild expression in the tumor tissue. (**c**) Negative TATI expression in the tumor tissue. Scale bar length 50 µm. Magnification ×200.

**Table 1 cancers-13-02811-t001:** Clinicopathological data according to S-TATI.

Variable	S-TATI−	%	S-TATI+	%	*p*-Value	Missing/% (*n* = 90)
Number of patients	69	76.7	21	23.3		
Age	60.8		65.1		0.054	
Gender						
Male	52	78.8	14	21.2		
Female	17	70.8	7	29.2	0.4	
Smoking						
Never	27	96.4	1	3.6		
Former	25	83.3	5	16.7		
Current	17	53.1	15	46.9	<0.001 **	
Heavy alcohol use						16/17.8
Never	36	83.7	7	16.3		
Former	6	60.0	4	40.0		
Current	13	61.9	8	38.1	0.09	
T class						
T1-T2	46	51.1	12	13.3		
T3-T4	23	25.6	9	10.0	0.4	
N class						
N0–N1	59	79.7	15	20.3		
N2–N3	10	62.5	6	37.4	0.2	
Stage						
I-II	52	82.5	11	17.5		
III-IV	17	63.0	10	37.0	0.04 *	
Grade						
I	1	33.3	2	66.7		
II	8	53.3	7	46.7		
III	60	83.3	12	16.7	0.009 *	
Localization						
Tonsil	48	90.6	5	9.4		
Base of tongue	17	77.3	5	22.7		
Soft palate	3	30.0	7	70.0		
Posterior wall of oropharynx	1	20.0	4	80.0	<0.001 **	
HPV						
HPV−	21	56.8	16	43.2		
HPV+	48	90.6	5	9.4	<0.001 **	

Abbreviations: TATI: Tumor-associated trypsin inhibitor; HPV: Human papillomavirus. *p* < 0.05 *, *p* < 0.001 **.

**Table 2 cancers-13-02811-t002:** Multivariate Cox regression analysis for overall-survival (OS) and disease-specific survival (DSS).

Variable	OS			DSS		
	HR	95% CI	*p*-Value	HR	95% CI	*p*-Value
Age	1.08	1.03–1.13	0.004 *	1.07	1.01–1.14	0.018 *
Smoking			0.034 *			0.205
Ex-smoker versus never	1.20	0.27–5.40	0.816	0.62	0.10–3.98	0.615
Current smoker versus never	4.14	1.22–14.06	0.023 *	2.43	0.64–9.18	0.192
Stage III–IVversusStage I–II	1.71	0.70–4.20	0.243	2.19	0.72–6.64	0.168
HPV-versus HPV+	1.01	0.36–2.83	0.988	0.96	0.27–3.43	0.955
S-TATI	2.47	1.26–4.84	0.009 *	2.54	1.07–6.02	0.034 *

HR: Hazard ratio; CI: Confidence interval. S-TATI values are log-transformed. *p* < 0.05 *.

**Table 3 cancers-13-02811-t003:** Clinicopathological data according to TATI expression in tumor tissue.

Variable	TATI in Tumor 0–1	%	TATI in Tumor 2–3	%	*p*-Value	Missing/% (*n* = 77)
Number of patients	20	26.0	57	63.3		
Age	64.4		61.1		0.2	
Gender						
Male	15	25.4	44	74.6		
Female	5	27.8	13	72.2	0.8	
Smoking						
Never	8	33.3	16	66.7		
Former	7	28.0	18	72.0		
Current	5	17.9	23	82.1	0.4	
Heavy alcohol use						14/18.2
Never	14	35.9	25	64.1		
Former	1	16.7	5	83.3		
Current	4	22.2	14	77.8	0.4	
T class						
T1–T2	11	22.4	38	77.6		
T3–T4	9	32.1	19	67.9	0.4	
N class						
N0–N1	12	19.0	51	81.0		
N2–N3	8	57.1	6	42.9	0.006 *	
Stage						
I–II	9	17.0	44	83.0		
III–IV	11	45.8	13	54.1	0.007 *	
Grade						
I	0	0.0	2	100.0		
II	3	23.0	10	77.0		
III	17	27.4	45	72.6	0.7	
Localization						
Tonsil	10	21.7	36	78.3		
Base of tongue	6	35.3	11	64.7		
Soft palate	3	33.3	6	66.7		
Posterior wall of oropharynx	1	20.0	4	80.0	0.7	
HPV						
HPV−	9	29.0	22	71.0		
HPV+	11	23.9	35	76.1	0.6	

Abbreviations: TATI: tumor-associated trypsin inhibitor; HPV: Human papillomavirus, TATI immunoexpression was scored in the tumor tissue, TATI in tumor 0–1: mild positivity, TATI in tumor 2–3: moderate-strong positivity, *p* < 0.05 *.

**Table 4 cancers-13-02811-t004:** Clinicopathological data according to TATI expression in tumor-adjacent lymphocytes.

Variable	TATI in Lymphocytes 0–1	%	TATI in Lymphocytes 2–3	%	*p*-Value	Missing/% (*n* = 76)
Number of patients	30	39.5	46	60.5		
Age	62.6		61.3		0.6	
Gender						
Male	23	39.7	35	60.3		
Female	7	38.9	11	61.1	0.9	
Smoking						
Never	10	43.5	13	56.5		
Former	5	20.0	20	80.0		
Current	15	53.6	13	46.4	0.04 *	
Heavy alcohol use						14/18.4
Never	17	44.7	21	55.3		
Former	1	16.7	5	83.3		
Current	9	50.0	9	50.0	0.4	
T Class						
T1–T2	14	28.6	35	71.4		
T3–T4	16	59.3	11	40.7	0.009 *	
N Class						
N0–N1	23	36.5	40	63.5		
N2–N3	7	53.8	6	46.2	0.2	
Stage						
I–II	16	30.2	37	69.8		
III–IV	14	60.9	9	39.1	0.01 *	
Grade						
I	1	50.0	1	50.0		
II	6	46.2	7	53.8		
III	23	37.7	38	62.3	0.8	
Localization						
Tonsil	15	32.6	31	67.4		
Base of tongue	6	37.5	10	62.5		
Soft palate	6	66.7	3	33.3		
Posterior wall of oropharynx	3	60.0	2	40.0	0.2	
HPV						
HPV−	18	60.0	12	40.0		
HPV+	12	33.3	34	66.6	0.003 *	

Abbreviations: TATI: Tumor-associated trypsin inhibitor; HPV: Human papillomavirus, TATI immunoexpression was scored in the inflammatory cells adjacent to the tumor tissue, TATI in lymphocytes 0–1: mild positivity, TATI in lymphocytes 2–3: moderate-strong positivity, *p* < 0.05 *.

## Data Availability

The data presented in this study are available on request from the corresponding author. The data are not publicly available due to patient data security.

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
