# Peer review of "Tumor-Associated Trypsin Inhibitor (TATI) as a Biomarker of Poor Prognosis in Oropharyngeal Squamous Cell Carcinoma Irrespective of HPV Status"

_cancers, 2021, doi:10.3390/cancers13112811_

Round 1
Reviewer 1 Report
This is an interesting study by Sjöblom et. al. proposing a serological biomarker to stratify OPSCC patients. As current de-escalation trials failed using p16 as a biomarker, this is an interesting study. However, there are limitations:
Major
- Please confirm the results using an independent validation cohort.
- What is the evidence of specificity of the assay towards detection of “TATI”?
- Please provide a multivariate analysis, adding also information on p16 and HPV-DNA as single features. This is important, as it has been shown that combination of both markers as dichotomous biomarkers (resulting in four possible states: ++/+-/-+/--), are informative.
- While the quantification of “TATI” using tumor tissues seems interesting, I am afraid it is highly questionable to use an antibody that detects soluble forms of a given molecule and to then propose sophisticated quantification of this. Either provide an additional quantitative measurement of TATI (ISH, qPCR) or discuss the (major) limitations of this approach and refer to future validation necessity.
Minor
- Abstract: “[…] We aimed to study the role of […]” should be “[…] We studied the role of […]”
- Abstract: “[…] In HPV- negative disease […]” should be “[…] In HPV-negative cases […]”
- Abstract: “[…] was linked with poor survival […]” should be “[…] was linked to poor survival […]”
- Introduction, second paragraph: Please cite primary (more and recent) literature on (novel) biomarkers to stratify OPSCC patients.
- Which treatment did the individuals receive?
Author Response
REVIEWER #1
Major Revisions
Comment 1: Please confirm the results using an independent validation cohort.
Response 1: We would like to thank the Reviewer for the professional feedback. Unfortunately, we are unable to use an independent validation cohort, as the serum samples for TATI quantifications were collected prospectively for the current study, and this procedure is not routinely performed in the management of OPSCC in Finland. In the future we will possibly have an opportunity to use biobanked samples from another cohort to perform a validation.
Comment 2: What is the evidence of specificity of the assay towards detection of “TATI”?
Response 2: We thank the Reviewer for this comment. The specificity of the antibodies for TATI has been demonstrated by mass spectrometry (Ravela S, Valmu L, Domanskyy M, Koistinen H, Kylanpaa L, Lindstrom O, et al. An immunocapture-LC-MS-based assay for serum SPINK1 allows simultaneous quantification and detection of SPINK1 variants. Anal Bioanal Chem. 2018;410(6):1679-88. We have included this reference in our study and in the revised manuscript. For addition, please see lines 146 and 449-451.
Comment 3: Please provide a multivariate analysis, adding also information on p16 and HPV-DNA as single features. This is important, as it has been shown that combination of both markers as dichotomous biomarkers (resulting in four possible states: ++/+-/-+/--), are informative.
Response 3: This is an important point and we thank the Reviewer for correctly emphasizing it. A multivariable analysis has now been performed and included in the final study and the results are presented in Table 2. Additions have been made in the manuscript (Materials and Methods, Results and Discussion). For these additions, please see Table 2. and lines 179-182, 202-210 and 270-272.
We believe it is not warranted to include p16 and HPV DNA in the analysis as single features. It has been shown that p16 is only a surrogate marker and is not necessarily associated with HPV positivity in all cases and thus, p16 as a biomarker lacks specificity in HPV detection. Additionally, combining p16 and HPV DNA with Smeets’ algorithm improves the specificity of HPV without diminishing the sensitivity as described in our references (#32 Smeets SJ et al, 2007), lines 437-439
Comment 4: While the quantification of “TATI” using tumor tissues seems interesting, I am afraid it is highly questionable to use an antibody that detects soluble forms of a given molecule and to then propose sophisticated quantification of this. Either provide an additional quantitative measurement of TATI (ISH, qPCR) or discuss the (major) limitations of this approach and refer to future validation necessity.
Response 4: We thank the Reviewer for pointing this out. The immunohistochemistry (IHC) was performed with a monoclonal antibody prepared as described in our “Materials and Methods” section. We are well aware of the potential specificity problems, however, the problems are caused by lack of reactivity of monoclonal antibodies (MAbs) with TATI after formalin fixation and paraffin embedding of tissue sections. This treatment often destroys the 3-D structure of proteins. Most monoclonal antibodies recognize the 3-D structure of proteins and peptides and polyclonal antisera usually contain some antibodies that recognize antigens in FFPE treated tissues. We have produced 20 monoclonal antibodies to TATI and were thus able to select MAbs that recognize TATI in FFPE-treated tissues. We have used TATI-MAbs for staining of various tissues and have not found evidence of nonspecific staining.
This clarification has been added capsulized in the revised manuscript. For the addition, please see lines 159-161.
Minor revisions
Comment 5: Abstract: “[…] We aimed to study the role of […]” should be “[…] We studied the role of […]”
Response 5: We thank the Reviewer for pointing this out. For correction, please see line 34.
Comment 6:. Abstract: “[…] In HPV- negative disease […]” should be “[…] In HPV-negative cases […]”
Response 6: We thank the Reviewer for pointing this out. For correction, please see line 42.
Comment 7: Abstract: “[…] was linked with poor survival […]” should be “[…] was linked to poor survival […]”
Response 7: We thank the Reviewer for pointing this out. For correction, please see line 43.
Comment 8: Introduction, second paragraph: Please cite primary (more and recent) literature on (novel) biomarkers to stratify OPSCC patients.
Response 8: We thank the Reviewer for this valuable note. For the additions, please see lines 69 and 393-398.
Comment 9: Which treatment did the individuals receive?
Response 9: This is an important point and we thank the Reviewer for the note. The patients received either (chemo)radiotherapy or surgical treatment with or without post-operative (chemo)radiotherapy. We have added information about the treatment in the revised manuscript (sections Materials and Methods). For this revision, please see lines 105 -107.
End of authors' response to Reviewer #1
Reviewer 2 Report
Interesting and well written m/s that investigates the potential prognostic value of a biomarker: Tumour associated trypsin inhibitor (TATI), for the outcome of patients with oropharyngeal cancer (OPC). OPC can be a very morbid cancer and the incidence is rising; new technologies that can help in its management are therefore welcome.
Thank you for the opportunity of conducting this review
I have some queries about the manuscript which could benefit from address
More exposition on where the 17 ug cut of for TATI positivity in serum would be helpful (see line 174)
My main issue with the m/s is around the analysis. While it is interesting that the OS curves show that serum TATI (negative) status appears to be associated with improved outcome in the HPV positive AND HPV negative cohort the denominators are quite/very small, particularly the HPV negative cases. Also I am not sure if the analysis performed allows for any interactions between the different variables to be assessed? Is it possible to perform a multivariate analysis which could help determine what factors (if any) in this cohort are independently associated with improved outcome. Playing “devils advocate” if, for example – TATI status is simply a surrogate for one or two variables that are routinely collected (such as HPV status and stage) what is the added value?
End of review
Author Response
First of we would like to thank the Reviewer #2 for this professional feedback, that allowed us the improve the manuscript significantly.
Please, find our response underneath.
Comment 1: More exposition on where the 17 ug cut of for TATI positivity in serum would be helpful
Response 1: We thank the Reviewer for this useful comment and this issue is now acknowledged. The reference range was determined on the basis of the central 95% reference interval in a group of 152 apparently healthy subjects. The lower reference limit was 3.1 µg/L and the upper reference limit 16 µg/L. Setting the cut-off value above the upper reference limit is usually recommended. The method is described in our references (#34 Osman et al., 1993, lines 443-445). We have highlighted this in the revised manuscript, for addition please see lines 141-142 and 146.
Comment 2: My main issue with the m/s is around the analysis. While it is interesting that the OS curves show that serum TATI (negative) status appears to be associated with improved outcome in the HPV positive AND HPV negative cohort the denominators are quite/very small, particularly the HPV negative cases. Also I am not sure if the analysis performed allows for any interactions between the different variables to be assessed? Is it possible to perform a multivariate analysis which could help determine what factors (if any) in this cohort are independently associated with improved outcome.
Response 2: We appreciate that the Reviewer #2 pointed out the issue and we agree with this note. Univariable and multivariable analysis have been included in the study and the Cox regression multivariate model is presented in Table 2. The multivariate analysis has now been noted in the following sections; Materials and Methods, Results and Discussion, respectively. For these additions, please see Table 2. and lines 179-182, 202-210 and 270-272
Comment 3: Playing “devils advocate” if, for example – TATI status is simply a surrogate for one or two variables that are routinely collected (such as HPV status and stage) what is the added value?
Response 3: We are thankful for this valuable comment. We believe that the possibility of obtaining prognostic information from a serum sample forms a definitive added value for TATI determination as currently there does not exist any serum-based determination possibilities for OPSCC. We have brought this up in the following sections; Introduction, Discussion (please see lines 72-77 and 330-332).
Reviewer 3 Report
The paper is very interesting and demonstre the possibiliti to use TATI as prognostic marker in HPV+ and - oropharyngeal squamous cell carcinoma.
i think it can be published in this form
Author Response
Reviewer #3
Comment 1: The paper is very interesting and demonstre the possibiliti to use TATI as prognostic marker in HPV+ and - oropharyngeal squamous cell carcinoma.
i think it can be published in this form
Response 1: We sincerely thank the Reviewer #3 for this feedback.
Round 2
Reviewer 1 Report
Thank you for addressing my comments.
Please provide the p16-status for the samples to your multivariate analysis, as this is part of the current tumor staging applied in daily clinical practice. (Please refer to my previous comment, although I scientifically agree that only cases with both HPV-DNA and p16-positivity are “truly” HPV dependent). P16/HPV combination indeed identifies subgroups (Rasmussen JH, Grønhøj C, Håkansson K, Friborg J, Andersen E, Lelkaitis G, Klussmann JP, Wittekindt C, Wagner S, Vogelius IR, von Buchwald C. Risk profiling based on p16 and HPV DNA more accurately predicts location of disease relapse in patients with oropharyngeal squamous cell carcinoma. Ann Oncol. 2019 Apr 1;30(4):629-636. doi: 10.1093/annonc/mdz010. PMID: 30657857)
While my comment on quantifying a soluble molecule with help of IHC is specifically questioning the methodical approach to quantify a protein that is “released” from cells to paracellular spaces, the authors provide a comment on antibody specificity (please see my previous comment). Methodically, how would quantifying a molecule that is released by cells and stained by the antibody by normalized using a given quantification method? In the event the authors propose a specific staining of TATI using their antibody, what compartments (cytoplasm, nucleus) would the authors think their antibody is staining? Please provide a larger overview for at least one case to give an overview of the staining to the reader. Please add scale bars to the image crops.
Please remove the last paragraph from the Discussion "Authors should discuss the results and how they can be interpreted from the per-333 spective of previous studies and of the working hypotheses. The findings and their impli-334 cations should be discussed in the broadest context possible. Future research directions 335 may also be highlighted." (page 12)
Author Response
The authors thank the Reviewer #1 for the professional feedback as well as for the earlier contributions.
Comment 1: Please provide the p16-status for the samples to your multivariate analysis, as this is part of the current tumor staging applied in daily clinical practice.
Response 1: We thank the Reviewer for this comment and it has now been addressed in the revised version of the manuscript. We have now performed a multivariable analysis with p16 status as a separate variable and the results are presented in Supplemental Table 2. For these additions, please see Supplemental Table 2. and lines 207-210 and 344-345)
Comment 2: Methodically, how would quantifying a molecule that is released by cells and stained by the antibody by normalized using a given quantification method? In the event the authors propose a specific staining of TATI using their antibody, what compartments (cytoplasm, nucleus) would the authors think their antibody is staining? Please provide a larger overview for at least one case to give an overview of the staining to the reader.
Response 2: We thank the Reviewer for this helpful specification. We have now acknowledged this issue. The specificity of the antibody is described in the following study; Paju A, Hotakainen K, Cao Y, Laurila T, Gadaleanu V, Hemminki A, et al. Increased expression of tumor-associated trypsin inhibitor, TATI, in prostate cancer and in androgen-independent 22Rv1 cells. Eur Urol. 2007;52(6):1670-9. At this moment, we are unfortunately unable to accurately point out which compartments of the TATI-molecule the antibody stains. It has been shown that the cytoplasm is the compartment stained in cells, and this type of TATI-staining has been shown in other tumors such as bladder, colorectal and renal cancer and thus, the method and the specificity have been thoroughly validated. (Please see our references #34 lines 444-446, #37 lines 453-454 and #43 lines 467-469). We have added clarification in the “Materials and Methods” and “Results” sections and the reference mentioned earlier (Paju et al. 2007) have been included in the revised manuscript. For these additions, please see lines 153, 159-161, 225-226 and 453-454)
Comment 3: Please add scale bars to the image crops.
Response 3: We thank the Reviewer for this note. Scale bars have now been added to the image crops. Please see Figure 2 and lines 239-241.
Comment 4: Please remove the last paragraph from the Discussion "Authors should discuss the results and how they can be interpreted from the per-333 spective of previous studies and of the working hypotheses. The findings and their impli-334 cations should be discussed in the broadest context possible. Future research directions 335 may also be highlighted." (page 12)
Response 4: We appreciate that the Reviewer pointed out this error. A correction has been made in the revised manuscript.
END OF AUTHORS' RESPONSE TO THE REVIEWER #1
Reviewer 2 Report
The authors have improved the manuscript through their comprehensive address of reviewer comments.
Author Response
Comment 1: The authors have improved the manuscript through their comprehensive address of reviewer comments.
Response 1: We thank the Reviewer #2 for this feedback, as well as for the earlier contributions.